# Trauma, Power, and Psychological Safety: Understanding the Mental Health Impact of Workplace Bullying

**DOI:** 10.3390/healthcare13233084

**Published:** 2025-11-27

**Authors:** Jason Walker

**Affiliations:** Departments of Health and Wellness Psychology and Industrial-Organizational Psychology, Adler University, Vancouver, BC V6B 3J5, Canada; jwalker4@adler.edu

**Keywords:** workplace bullying, psychological safety, trauma, power, leadership, organizational justice, occupational health

## Abstract

**Highlights:**

**What are the main findings?**
Workplace bullying, harassment, and sexual abuse are strongly associated with anxiety, depression, PTSD, burnout, sleep disruption, cardiovascular problems, absenteeism, and turnover.Diminished psychological safety and toxic or abusive leadership styles intensify psychological harm, while ethical, inclusive, and trauma-informed leadership reduces it.A strong psychosocial safety climate, combined with trauma-informed approaches, mitigates re-traumatization, restores trust, and supports recovery.The most effective organizational response is multi-level, integrating policy, leadership accountability, safety climate, and targeted supports.

**What are the implications of the main findings?**
Workplace bullying should be recognized and addressed as a form of violence and a public and occupational health hazard requiring urgent prevention and intervention.Embedding psychological safety as a core organizational value is essential to reducing harm, strengthening resilience, and improving workforce retention and well-being.Leaders must adopt trauma-informed and ethical practices to prevent recurrence, rebuild trust, and shape healthier organizational cultures.

**Abstract:**

Background: Workplace bullying, harassment and sexual abuse cause psychological harm, and can pose a significant threat to the success of an organization as well. This type of violence in the workplace, comprising negative actions and often abuse of power, can lead to trauma, anxiety, depression, PTSD and in severe cases, suicide. These acts impact workplace performance, negatively impact psychological safety and lead to high turnover and loss of productivity in an organization. Objectives: This narrative review outlines the key concepts of bullying, its impact on the individual, and the ways an organization can obstruct and manage it, using recent works (2018–2025) and some highlighted literature on trauma, power, and psychological safety. Methodology: Research conducted on leadership, safety climate, psychological safety and trauma-informed- as well as meta-analyses and relevant gray literature, journal articles, and other studies on bullying that A narrative synthesis of peer-reviewed and selected gray literature was conducted across PsycINFO, MEDLINE, Scopus, and Web of Science were integrated to this review. Results: Exposure to bullying was connected to anxiety, depression, burnout, post-traumatic stress disorders, cardiovascular problems, absenteeism, and turnover. Diminished psychological safety, as well as disordered leadership, increases the damaging effect. In contrast, ethical trauma-informed leadership and a strong psychosocial safety climate promote recovery and decrease the incidence of bullying. Conclusions: Recognizing workplace bullying, harassment, and sexual abuse as forms of violence—and as both occupational and public health hazards—underscores the urgency of prevention. Embedding psychological safety as a core organizational value at every level is essential to fostering healthier, more resilient workplaces.

## 1. Introduction

Workplace bullying, harassment, and sexual abuse are characterized as unwanted negative actions directed at an employee who is unable to defend themselves that occur repeatedly [1,2]. This form of violence is associated with deep individual harm, as well as organizational harm, within the workplace [3,4]. The definition of bullying includes an array of actions that an individual can take, such as overt actions of ridicule, verbal abuse, and threats with the covert actions of social exclusion, psychological terror, sabotage, and workload manipulation [1,3,5]. Moreover, it is easy to differentiate bullying from normal workplace conflict in terms of persistence, intent to do harm, and an imbalance of power in favor of the perpetrator [1,6]. Recent surveys and probability samples suggest that approximately one in ten employees experience bullying in the workplace in a given year, with some estimates suggesting even higher numbers, especially in certain industries and with broader definitions of workplace bullying [2,7]. Having to deal with bullying in the workplace is a mental health concern, and therefore needs to be categorized and elevated beyond a simple problem for Human Resources, to the whole organization and beyond, as a matter of public health concern [4,8,9].

Those who are targeted have a correlation with heightened levels of anxiety, the tendency to burn out, and developing depressive symptoms; a subset of the bullied can be considered to be psychologically traumatized, with a portion even describing suicidal behavior or intent [4,5]. Prolonged or chronic activation of the body’s stress systems has also been shown to be related to risks of sleep disturbances and cardiometabolic and more [6,7]. Prevalence research indicates that roughly one in ten employees experience workplace bullying each year, with substantially higher rates in healthcare, education, and emergency-service sectors where interpersonal stress and hierarchical pressure are greatest.

For organizations, the consequences of bullying behavior translate to absenteeism, lower productivity of actually present employees, more intent to leave, withdrawal from work, and even diminishing engagement and morale within a workgroup or team [7]. These consequences have significant costs on both a societal and an organizational level. To understand the reasons for such extensive damage from bullying behavior, it is necessary to examine much more carefully the neglect of the interplay between power and trauma, safety, or, more accurately, the absence of it. Psychological safety is the shared belief within a team that taking interpersonal risks—such as admitting mistakes, asking for help, or reporting misconduct—will not lead to punishment or humiliation [8,9]. Having a safe work environment is an essential protective factor; when psychological safety is high, both targets and witnesses feel empowered to speak up, leaders are receptive, and issues can be resolved before they escalate [8,9]. Conversely, when psychological safety is weak, employees often remain silent, allowing abuse and misconduct to continue unchecked [9]. This review draws on Lazarus and Folkman’s [10] stress–appraisal model and Herman’s [11] trauma framework to conceptualize workplace bullying as a chronic relational stressor capable of eliciting trauma-like responses. Positioning bullying within these models frames it as both an occupational and public-health issue.

This review focuses on collecting the current evidence and practice guidance to (a) define bullying through and trauma and stress appraisal theory [12,13,14], (b) clarify the psychological and physiological consequences [4,5,6,7], (c) examine the leadership, power, and organizational context [15,16,17,18,19,20,21,22], and (d) describe multi-level prevention and intervention approaches that focus on psychological safety and trauma-informed leadership [19,21,22,23,24,25]. Foundational and seminal contributions are retained, and recent empirical work is stressed to inform current practice.

## 2. Materials and Methods

This study employs a narrative review approach to synthesize empirical, theoretical, and applied literature on workplace bullying, trauma, and psychological safety. The databases PsycINFO, MEDLINE, Scopus, and Web of Science Motor were searched from 2018 to 2025 using the keywords workplace bullying, harassment, and mobbing, psychosocial trauma, PTSD, leadership chains, and destructive, ethical and transformational leadership, organizational justice, psychosocial safety climate, and intervention. The inclusion criterion emphasized peer-reviewed empirical articles, systematic reviews, and meta-analyses on the following topics: the outcomes of bullying, and antecedents and moderators such as leadership, justice, safety climate, and interventions. Conceptual foundational literature such as theories and works on trauma and stress and the trauma model and stress-appraisal frameworks were also included. Some gray literature such as practitioner essays and Forbes columns of the corresponding author was included to show applied practice and current discourse on which peer-reviewed evidence is still emerging. The selected data were thematically analyzed and organized into four groups: trauma and stress-appraisal, psychological and physiological consequences, leadership in organizational context, and psychological safety as the primary focus of prevention and intervention. Searches were conducted in PsycINFO, MEDLINE, Scopus, and Web of Science for studies published between 2018 and 2025 using the keywords ‘workplace bullying’, ‘harassment’, ‘mobbing’, ‘psychosocial safety climate’, ‘trauma’, ‘leadership’, and ‘organizational justice’. A total of 68 sources were reviewed. Inclusion criteria included peer-reviewed empirical studies, meta-analyses, and theoretical works; exclusion criteria comprised non-English texts, duplicates, and non-relevant publications. Gray literature was consulted only to address gaps in empirical evidence. Peer-reviewed research was prioritized, and each study was qualitatively appraised for rigor, recency, and relevance. Four themes emerged: (1) trauma and stress-appraisal, (2) psychological and physiological consequences, (3) leadership and organizational context, and (4) prevention and intervention through psychological safety.

## 3. Results and Discussion

### 3.1. Trauma and Stress-Appraisal Frameworks

Bullying behavior impacts the target similarly to a traumatic event, often resulting in chronic, severe, relational stress, and relational post-traumatic-like outcomes [4,12,13]. Each individual responds differently; however, most experience varying degrees of shame, avoidance, and hypervigilance, alongside an inability to trust, demonstrating the complex trauma paradigm as exposure grows [12]. Stress appraisal theory clarifies this kind of behavior: when faced with constant hostility, the victim feels trapped because manipulation, aggression, and power differentials (like the authority figure in these cases) come into play [13]. The individual eventually stops trying to overcome and feels as though escape is impossible [12,13]. The target often describes the circumstances where they feel unexplainably socially trapped, experience constriction of freedom, stress, and unreasonable aggression, explainable panic (extreme neural hyperarousal), where the system of control gets triggered and depleted [6,12]. A prolonged state of dysregulation of the stress response attempts to control homeostatic state equilibrium. The HPA axis loses effective control of the reset mechanisms during the sleep cycle, along with obsessive precursors and self-induced pseudo-faced anxiety [6,7,12]. Such circumstances expose, in copious amounts, stress and excessive diurnal strain [6,7]. The individual enters a stress and performative dysfunction cycle, as they feel this stress and continue to operate dysfunctionally over prolonged periods [6,12,13].

Bullying quite often results in moral injury when those in authority do something unethical, or when organizational responses are dismissive or retaliatory, concern, and betray the expectation. “Moral injury is the betrayal of what is right by one who has legitimate authority in a socially significant setting, and who triggers shame, anger, and demoralization” [14]. In organizational settings, it is when a leader abuses or when an organization intentionally acts to minimize harm to itself. Such institutional betrayal deepens the trauma, diminishes the sense of purpose in the workplace, and hinders the process of healing [20,21]. Practicing trauma-informed strategies, such as prioritizing the safety, trustworthiness, and empowerment of individuals, speaks to the wounds of injury by recognizing the injury, avoiding re-traumatization, and regaining lost control and authority over one’s life [12,21].

### 3.2. Psychological and Physiological Outcomes

Across various designs and geographical regions, the prevalence of bullying is linked with anxiety, depression, and burnout, and longitudinal studies substantiate bullying as a precursor to the decline of one’s mental health [3,4]. Meta-analytic studies demonstrate moderate to high associations with distress and PTSD-related symptoms [4]. Although not all the time, suicidal ideation and attempts are higher amongst the most targeted individuals [5]. From a physiological standpoint, chronic bullying is linked with and increases the risk for cardiovascular disease, as well as contributes to sleep disturbances and other stress-related symptoms [6,7]. In a multicohort European study, chronic exposure to workplace bullying was found to predict the development of heart disease, independent of baseline risk factors, suggesting chronic psychological stress as a causal pathway [6].

Bullying, at the organizational level, lowers employee productivity caused by presenteeism and absenteeism, decreases motivation, and increases employee turnover, particularly amongst high performers [7,10]. Employees disengaged in collaborative work self-protect and withdraw to avoid interpersonal risk, leading to a decline in team cohesion and success. Worse, the harm can be amplified by the bystanders, e.g., witnesses of bullying who, in turn, suffer climate harassment, stress, moral dilemmas, and the risk of retaliation, despite not being the targets. The implications on the targets, bystanders, and teams amplify the argument of bullying being treated as a public and organizational health problem, rather than a private interpersonal issue [1,2,3].

### 3.3. Leadership and Organizational Context

In cases of bullying at the workplace, leadership acts as a starting point as well as an escalator of the dynamics. Aggressive and domineering leadership or the absence of it ‘leave alone’ leadership typologies normalize the absence of aggression and advocate lawlessness as a rule ‘free aggression’ [16]. In bullying cases, the concentration of leaders with certain destructive behavioral patterns marked by dominance, callousness, and lack of empathy. These leaders, along with their subordinates, engage in numbness and gaslighting, which the subordinates do not challenge as a show of surrender [17]. While these leaders often achieve short-term targets that may appear as “higher performance,” upper management tends to reward their actions in an implicit or structural (“built-in”) manner, which inadvertently reinforces abusive behavior [16,17]. In contrast, ethical and transformational leadership styles are associated with lower bullying prevalence, as they model respect, fairness, and accountability, foster trust, and amplify employee voice [18,19]. While downward bullying from supervisors remains most frequent, lateral (peer-to-peer) and upward (subordinate-to-supervisor) bullying also occur, illustrating the multidirectional nature of power abuse in organizations.

Within the psychosocial predictors of individual bullying and well-being generalized across workgroups, there exists a distinct psychosocial domain of safety—known as the psychosocial safety climate—which serves as a strong predictor of both well-being and exposure to bullying [22]. A high degree of psychosocial safety at work (PSC) guarantees the absence of workplace aggression and better overall workplace ‘bully-proof’ policies. In low psychologically protected (PSC) environments, the absence of workplace aggression, bullying, and victimization cultivates ‘pro target policies, which equate to an empty promise with the absence of ongoing, nefarious workplace aggression [20]. Organizational power differentials, which are always present, could, if properly aligned, along with other nefarious practices such as unchecked multi-channel reporting, independent investigations, and other retaliatory practices, ‘bring to heel’ organized systemic aggression [18,19,20,22].

Psychological safety, especially at the team level, which is significantly influenced by the actions of the team leader, mitigates risk by offering the twin capabilities of voice and protection at the most basic level [8,9,22,26,27]. Leaders who appreciate feedback, invite suggestions, and describe potential issues as collective learning events enhance the willingness of team members to voice concerns across a wide spectrum of issues, even misconduct, without retribution [8,9,28,29,30]. In teams that enjoy psychological safety, it is challenging for aggressors to single out and target the bullied. Colleagues counter harmful stereotypes and, too, engage leadership or other formal processes earlier, reducing the interval and intensity of exposure [8,9,22]. In teams that lack psychological safety, the default is silence, which is permissive to prolonged exposure and escalation [9,22,30].

### 3.4. Prevention and Intervention

The most effective solutions are those that are multidimensional and combine policy, the capability of leadership, climate, and relevant supports. Primary prevention (Organizational systems): Organizations should document anti-bullying policies and provide definitions, examples, anti-bullying sanctions, anti-retaliation policies, and provisions that are strong, and which administrative and legislative enforcement policies contain (transparent reporting systems, independent hotlines, ombudspersons, and trained investigators) [18,19,20]. Policies should be unbundled from policies and implemented in policies regarding leadership and accountability. Civility, just treatment, and immediate response to complaints should be embedded in leader selection, onboarding, and performance review in policies. Strong PSC should be built from psychosocial risk management in health and safety (integration of ISO 45003) and climate assessment from reviewed and routinely monitored confidential surveys [22]. Feedback loops from executives and managers (dashboards) foster ongoing improvement and are captured as data-driven.

Practices and advancement of leadership: Ethical and trauma-informed skill development capability frameworks that a program seeks to build on include: listening with empathy, role model and respect, emotion regulation, non-punitive feedback, and early intervention on incivility [21,23]. Indicators of trauma (e.g., hypervigilance, withdrawal) must be acknowledged, and the possibility of re-traumatization side-stepped, and reasonable accommodations (e.g., adjusted reporting lines during investigations) co-designed. Embedding Psychological Safety at the team level through short modules on speaking up, setting up consultation norms, and appreciation demonstrated the ability to decrease bullying and supervisor intimidation while sustaining engagement in cluster-randomized trials [25]. Teams employ practices to institutionalize safety and incorporate practices like structured debriefs after risky incidents, rotating facilitation, and explicit norms for safe dissent.

Secondary interventions (response and support): The onset of the bullying behaviors or receiving the report of the behaviors should result in immediate and non-biased interventions. Separation of the individuals involved, changes in reporting procedures, or exculpating leave for the alleged perpetrators of the bullying, pending the determination of the facts, may constitute the preliminary measures. Targeted individuals should be provided with confidential counseling, trauma-focused psychological first aid from trained peers, and other psychological support, more specialized from the HR/employees assistance programs [21,23]. In some cases, with appropriate skill in facilitation and consent, restorative conferencing can repair some of the harm caused, re-establish some of the agreements, and re-integrate the parties, but it is contraindicated in cases of severe power imbalance or some forms of personality pathology that compromise safety [24]. Managers need to carry out detailed follow-ups to ensure there is no retaliation and that there is monitoring of the reintegration process. Witnesses, too, need to be supported, and the active bystander training programs can enable peers to act in a safe manner [25,26].

Tertiary and continuous improvement: For some severe or prolonged cases of bullying, longer-term arrangements may be required, such as flexible scheduling, modifications to shift patterns or worksite, coordination of medical leave with other forms of support, and, where applicable, assistance with occupational injury claims to facilitate recuperation [21,23,24]. Organizations need to conduct after-action reviews (ensuring confidentiality is respected) to derive system lessons: concerning early warning indicators, gaps in the process, and the level of manager capability. Aggregated learned lessons and communicated positioning reinforce the document and culture shift. Over time, the frameworks for psychological safety and trauma-informed approaches become the mainstays, yielding collateral positive outcomes in learning, innovation, and retention [8,9,23,25].

Crossing the chasm by adopting and integrating practitioner-facing guidance and gray literature has been shown to expedite the adoption of the frameworks. Examples of ease frameworks include the recognition of the act of gaslighting and gossip as a social weapon, as well as the more covert impacts of toxic workplaces. These frameworks enable leaders and employees to articulate the frameworks as well as try to help in the mobilization of the needed support. These are not peer-reviewed indicators of the voicing science. These unreferenced sources put forth are, to put it lightly, heuristics that untangle the science, and more importantly, flag the fact that organizations seem to have an abundance of untreated bullying as a non-negotiable, urgent, preventable hazard. Effective prevention is multidimensional, requiring leadership training, confidentiality protocols, and adequate resources. Adopting ISO 45003 psychosocial risk-management standards can strengthen organizational commitment to psychological safety.

### 3.5. Measurement and Assessment

Rigorous assessment is foundational for both prevention and accountability. Standardized tools such as the Negative Acts Questionnaire (NAQ) and its revisions remain widely used to quantify exposure to bullying across personal, task-related, and social dimensions, while team psychological safety can be assessed with validated short scales derived from foundational work [1,3,8,9]. Validated instruments such as the Negative Acts Questionnaire–Revised [1] offer robust measures for assessing bullying exposure and psychosocial risk. To build a high-fidelity picture, organizations should combine **multiple data sources**: periodic anonymous climate surveys, pulse checks after reorganizations, exit interviews coded for bullying indicators, and analysis of complaints and investigation outcomes. Because under-reporting is common, survey designs must protect anonymity and communicate **non-retaliation** policies credibly. Results should be disaggregated by unit, job level, contract type, and demographic categories to detect **hot spots** and inequities. From a systems perspective, tracking leading indicators (e.g., rates of incivility observations, time-to-response on complaints) alongside lagging indicators (absenteeism, turnover) supports **predictive monitoring**. Where resources allow, advanced analytics can model how psychosocial safety climate variables predict outcomes like team engagement and voluntary turnover, enabling targeted interventions [22,23,25]. Importantly, assessment is not a one-off project but an element of continuous improvement integrated into governance (e.g., quarterly review by an ethics or safety committee).

### 3.6. Contexts: Sectors, Remote/Hybrid Work, and Equity

Context matters. **High-risk sectors** include healthcare, emergency services, education, hospitality, and tech—settings characterized by high workload, role ambiguity, hierarchical constraints, or intense customer demands [1,2,3,10,11]. Newcomers, contingent workers, and minority groups often face disproportionate risk due to lower power, reduced voice, or stereotyping. In **remote and hybrid** workplaces, bullying can manifest as exclusion from meetings, public shaming in group chats, unreasonable monitoring, or ‘silent treatment’ via ignored messages. Digital traces may facilitate documentation, but asynchronous communication can also **amplify misunderstandings**. Leaders must adapt psychological safety practices to virtual contexts—explicit turn-taking, clear expectations for response windows, shared agreements on feedback channels, and routine one-to-ones to detect early signs of distress [8,9,23]. Organizations should ensure equitable access to opportunities and information across on-site and remote staff, mitigating power imbalances that can otherwise breed resentment and covert aggression [22].

### 3.7. The Economic Case and Return on Investment

Addressing bullying is often framed as a moral imperative—which it is—but it also presents a compelling **business case**. Bullying increases absenteeism and presenteeism, undermines engagement, and accelerates turnover; each mechanism carries quantifiable costs [7]. Conservatively, replacing a skilled professional can cost 50–200% of annual salary, while chronic stress elevates healthcare expenditures and disability claims. By strengthening psychosocial safety climate and team psychological safety, organizations can realize downstream benefits in innovation, error reporting, and customer satisfaction [8,9,22,23]. Trials that build psychological safety have shown concurrent improvements in engagement and reductions in reported intimidation [25]. A practical ROI model includes: avoided legal expenses from disputes; reduced vacancy and onboarding costs due to lower turnover; productivity gains from reduced presenteeism; and quality gains from earlier risk escalation. When presented to executive stakeholders, this integrated model reframes prevention as **investment** rather than compliance overhead.

### 3.8. Limitations and Future Directions

As a narrative review, this study is subject to selection bias and does not include a structured quality assessment, which should be addressed in future systematic work despite progress, important knowledge gaps remain. Measurement heterogeneity complicates meta-analytic estimates; cross-cultural validations of bullying and psychological safety measures are ongoing [3]. Longitudinal and intervention studies are still fewer than desired; more **pragmatic trials** are needed to test scalable, low-burden team practices in diverse sectors [23,25]. Future work should illuminate **neurobiological pathways** linking sustained exposure to dysregulation (e.g., cortisol profiles, sleep architecture) and examine **intersectional risks**, including how gender, race, disability, and precarious employment interact with power structures to shape vulnerability and help-seeking [3]. Remote and hybrid modalities warrant targeted study. Finally, the field would benefit from implementation science approaches that specify **fidelity criteria** for psychological safety and trauma-informed leadership programs, enabling replication and benchmarking.

**Implementation playbook**. Translating principles into practice benefits from a structured, time-boxed roadmap:**First 30 days—Stabilize and signal**. Executive sponsorship is named; a concise statement declares bullying a psychosocial hazard and reaffirms non-retaliation. A cross-functional taskforce (HR, Legal, OH&S, DEI, Operations) inventories current policies, reporting channels, investigation capacity, and data. Immediate **risk controls** are introduced (e.g., safe reporting channel outside supervisory line; triage protocol).**Days 31–90—Design and equip**. Co-design a plain-language anti-bullying policy with examples and sanctions aligned to jurisdictional requirements; publish a **decision tree** for leaders and employees (what to do if you witness, experience, or receive a report). Establish standard operating procedures for intake, triage, investigation, and post-finding remediation, including **retaliation monitoring** for 90 days. Build a short **leader micro-curriculum**: recognizing early signs, responding without bias, documenting, offering support, and engaging independent investigators when conflict of interest exists [18,19,20,24]. Pilot team psychological safety practices (check-ins, norms for dissent) in two to three units to localize learning [8,9,25].**Days 91–180—Scale and hardwire**. Integrate bullying prevention metrics into quarterly business reviews (e.g., time-to-first response; investigation cycle time; employee perceptions of safety and fairness). Incorporate **behavioral expectations** into performance management and promotion criteria for leaders (civility, fairness, responsiveness). Launch active bystander training and a confidential peer support network to provide psychological first aid. Codify a **remediation/repair menu** (coaching, mediated agreements, reassignment, demotion, termination) tied to severity and patterns. Publish anonymized case learnings to reinforce norms and transparency.

**Investigation standards**. Investigations should be timely, independent, and trauma-informed: interviews conducted in psychologically safe settings; the right to a support person respected; communications clear about process and timelines; and **minimization of re-exposure** to the perpetrator during inquiries. Findings should apply consistent evidentiary thresholds and be documented with a rationale. Where allegations are substantiated, **proportionate outcomes** protect the workforce and deter recurrence. Regardless of findings, the organization should offer support to all parties, recognizing that participation in investigations can be stressful [20,21,24].

**Training and capability**. A minimalist but high-impact curriculum focuses on: (1) the science of psychosocial hazards and why bullying is a public health issue [1,2,3,4,5,6,7,22]; (2) leader communication micro-skills (curiosity, paraphrasing, appreciative responses) that cultivate psychological safety [8,9,23]; (3) trauma-informed behaviors that prevent re-traumatization (predictability, choice, transparency) [12,21]; (4) early resolution skills (naming behaviors, boundary setting, structured feedback); and (5) pathways for formal action. Blended delivery (short e-modules, live practice, leader nudges) reduces time away from work while sustaining behavior change. Pair training with **coaching** for leaders managing complex cases.

**Metrics and governance**. Establish a light-weight **governance cadence**: monthly operational reviews in HR/OH&S; quarterly executive reviews; and an annual board-level summary. Core indicators include prevalence of reported negative acts, time-to-response, time-to-closure, perceptions of fairness, psychological safety scores, and downstream outcomes (absenteeism, turnover, engagement). Qualitative indicators (themes from open comments; learning from after-action reviews) round out the picture. Publicizing progress—while protecting confidentiality—builds credibility and sustains momentum. Over time, organizations should see a **paradoxical trend**: an initial rise in reporting as trust increases, followed by declines in severe incidents as prevention takes hold [22,23,25].

## 4. Conclusions

Workplace bullying is a preventable manifestation of organizational power imbalance and unsafe culture. Embedding trauma-informed leadership, ethical accountability, and psychosocial safety climate practices within organizational systems is critical to reducing harm and promoting well-being. Like many societal phenomena, workplace bullying, harassment and sexual abuse are acts of violence that can be prevented. The behavior typically stems from the inability to balance power and a compromised level of psychological safety. It brings about mental and physical harm, breaks trust, and lowers productivity. Reviewing the available literature on the subject, there is a clear call to action: psychological safety and trauma-informed leadership should no longer be mere goals, but core requirements in organizational policy. This starts with tangible policy development and application, leadership responsiveness, psychosocial safety climate, and support available in a timely fashion to targets and bystanders. Bullying struggles to take root in environments where compassion is institutionalized and employee voices are protected. Even wounds that cause harm can be healed in such cultures. Future research should focus on the neural pathways underlying recovery and resilience, explore flexible work arrangements—particularly within hybrid and remote systems—and address targeted, intersectional risks to equity in prevention and response. No workplace should regard bullying as an inevitable or undeserved consequence of “how things are”. Doing so forfeits the opportunity to cultivate safer, more creative, and more resilient organizations. Preventing bullying is not only the right thing to do—it is also economically advantageous.

## Data Availability

No new data were created or analyzed in this study.

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
