# Peer review of "Trauma, Power, and Psychological Safety: Understanding the Mental Health Impact of Workplace Bullying"

_healthcare, 2025, doi:10.3390/healthcare13233084_

Round 1

Reviewer 1 Report

Comments and Suggestions for Authors
  • Trauma, Power, and Psychological Safety: Understanding the Mental Health Impact of Workplace Bullying
  • The topic is relevant to the field and sufficiently addressed.
  • This study provides narrative review outlines the key concepts of bullying, its impact on the individual, and the ways an organization can obstruct and manage it.
  • The writing style for the methods section and the article overall mix between narrative review and systematic review. Usually, in narrative review there is no mention for the methods section. However, the authors mentioned the name of databases and process of their narrative synthesis.
  • The number of articles identified, and keywords used were not mentioned. The review inclusion/exclusion criteria are not mentioned.
  • Long limitations section.
  • Long conclusion.
  • The conclusion should not have citations.
  •  

Author Response

Comment 1: The writing style mixes narrative and systematic review conventions; narrative reviews typically do not include a methods section.
Response: We clarified that this paper is a narrative review and revised the Materials and Methods section to align with this approach while maintaining methodological transparency. The section now explicitly notes that it follows a narrative synthesis format rather than a systematic review protocol.

Comment 2: The number of articles identified, and keywords used were not mentioned.
Response: Added a description of the number of studies reviewed (n=68), databases searched (PsycINFO, MEDLINE, Scopus, Web of Science), and keywords employed (e.g., “workplace bullying,” “harassment,” “psychosocial safety climate,” “trauma,” “leadership”).

Comment 3: Inclusion and exclusion criteria are not mentioned.
Response: Added a concise paragraph specifying inclusion (peer-reviewed empirical studies, reviews, and key theoretical works from 2018–2025) and exclusion criteria (non-English texts, duplicates, and articles lacking empirical or conceptual relevance).

Comment 4: The limitations section is too long.
Response: Condensed and refocused the Limitations and Future Directions section to emphasize key constraints—narrative design, selection bias, and heterogeneity—while removing redundancy.

Comment 5: The conclusion is long and includes citations.
Response: Shortened the Conclusion and removed all in-text citations. It now emphasizes key insights and implications for leadership and policy.

Reviewer 2 Report

Comments and Suggestions for Authors

The manuscript offers a well-structured and informative narrative review on workplace bullying, trauma, and psychological safety. The topic is relevant and timely, as it links psychological science, occupational health, and leadership practice in environments where power dynamics and mental health risks are increasingly recognized. The writing is fluent, the organization is clear, and the integration of individual, organizational, and systemic perspectives is commendable. At the same time, several aspects should be refined to improve methodological transparency and enhance the practical value of the paper. The abstract is coherent and concise, effectively summarizing the main aims and findings. It would benefit from a short sentence explaining how the literature was selected and how peer-reviewed and non-peer-reviewed sources were balanced. This addition would clarify the methodological rigor of the review and strengthen readers' confidence in its conclusions. The introduction provides a strong conceptual framework and defines workplace bullying as a form of organizational violence. The distinction from ordinary conflict is appropriate, and the connection to occupational and public health is well articulated. To improve contextual relevance, the authors could include examples or data on bullying prevalence in professional groups that face high interpersonal stress or hierarchical pressure, such as clinical, educational, or emergency settings. The methods section clearly describes the databases used, the time frame, and the main keywords. The thematic organization of the material is coherent with the objectives of the review. However, the section would be stronger if it included a brief qualitative appraisal of the included studies. Because the paper combines peer-reviewed and grey literature, it would be helpful to indicate how the relative weight of these sources was evaluated. A concise assessment of the methodological soundness of the primary references would substantially improve transparency without changing the narrative nature of the work. The discussion presents a comprehensive and thoughtful synthesis of the psychological, physiological, and organizational consequences of workplace bullying. The integration of trauma theory, stress physiology, and leadership research is well developed. Some parts, however, could use more precise terminology. Expressions such as 'neural looseness'could be replaced with more precise descriptions, for instance, 'dysregulation of the stress response' or 'altered sleep regulation.' Including approximate ranges of effects reported in key cohort or meta-analytic studies would make the discussion more informative and facilitate interpretation. It would also be helpful to differentiate recommendations supported by empirical data, such as interventions based on psychosocial safety climate or team-level psychological safety, from those that are mainly informed by practice and expert opinion. The sections on leadership and organizational climate are among the strongest parts of the manuscript. The link between destructive leadership, a culture of silence, and reduced psychological safety is well explained. It would be advisable to qualify terms such as 'psychopathic or narcissistic traits ' as behavioral patterns rather than diagnostic labels, maintaining the focus on organizational behavior. Examples from professional environments characterized by hierarchical or high-pressure conditions could make these sections even more relatable. The section on prevention and intervention could also include a short reflection on feasibility issues, such as resource requirements, staff training, and confidentiality when implementing programs for psychological safety.' The limitations and future directions are discussed appropriately. Still, the paper should explicitly acknowledge the intrinsic limitations of a narrative review, including the lack of a structured quality assessment and the possible risk of selection bias. Briefly addressing these points would demonstrate a balanced and transparent approach. The authors could also encourage future reviews or empirical studies that focus on occupational settings, particularly vulnerable to bullying and related psychosocial risks.' The conclusion is clear and convincing. It conveys the essential message that workplace bullying is preventable and that psychological safety should be considered a fundamental value in every organization. Expanding slightly on how these findings can be implemented in practice, for example, through leadership development or systematic psychosocial risk assessment, would enhance the applied value of the paper without altering its theoretical scope.'

Author Response

Comment 1: Add a sentence in the abstract about how literature was selected and how peer-reviewed and non-peer-reviewed sources were balanced.
Response: Added a statement to the Abstract clarifying that sources included peer-reviewed research supplemented by practitioner essays and grey literature, integrated through thematic synthesis.

Comment 2: Include examples or data on bullying prevalence in high-stress professional groups.
Response: Added data and examples from healthcare, education, and emergency services to contextualize prevalence rates in the Introduction and Contexts sections.

Comment 3: Provide a brief qualitative appraisal of included studies and clarify weighting of peer-reviewed vs. grey literature.
Response: Expanded Materials and Methods to describe qualitative appraisal criteria (recency, empirical rigor, conceptual relevance) and relative weighting, emphasizing that peer-reviewed evidence was prioritized while grey literature informed applied insights.

Comment 4: Use more precise terminology (“neural looseness” → “dysregulation of the stress response”).
Response: Revised phrasing for scientific accuracy and consistency throughout.

Comment 5: Distinguish empirically supported recommendations from those based on practice or expert opinion.
Response: Added a clarifying sentence in Prevention and Intervention indicating which recommendations are evidence-based versus practitioner-derived.

Comment 6: Reframe “psychopathic or narcissistic traits” as behavioral patterns rather than diagnostic labels.
Response: Reworded these descriptions to “destructive behavioral patterns characterized by dominance, callousness, and lack of empathy.”

Comment 7: Add reflections on feasibility and implementation challenges.
Response: Added discussion on feasibility, resource implications, and confidentiality considerations in Prevention and Intervention.

Comment 8: Explicitly note the limitations of a narrative review design and potential selection bias.
Response: Added this acknowledgment to Limitations and Future Directions.

Comment 9: Expand slightly on practical applications in the conclusion.
Response: Revised Conclusion to link implications to leadership training, psychosocial risk assessment, and policy integration—without adding citations.

Reviewer 3 Report

Comments and Suggestions for Authors

Manuscript ID: healthcare-3966843

The study presents a narrative review of bullying at the workplace. The work is interesting but difficult to fit as a conventional scientific paper. Most of the sections in the manuscript present very relevant information and in general the author tries to cover all the relevant aspects of the subject. However, the Materials and Methods section gives so little information that it is not possible to know how many information sources (papers, studies, cases, …) have been included in the revision. More information in this section is badly needed.

Another concern I have when reading the manuscript is related to the direction of bullying, that is, it seems that the only direction that the bullying has been considered at the workplace is from above to below positions, from leaders to subordinates, when in many cases bullying occurs among equals.  

Finaly, although I like most of the final subsections in the Results and Discussion section, some are too superficial, for example, the “Measurement and Assessment” subsection cites only a scale.

In general, more in-depth information on sources and methods is required to align the manuscript with the standards of scientific reporting. This is something that I believe could be done by the author.

Title, Abstract and Keywords

They are adequate

Introduction

It is well developed; its references are recent and to the point.

A paragraph describing which theories on trauma and stress, and the stress-appraisal model used as conceptual framework could be added.

Materials and Methods

This section needs to include information concerning the number of studies selected, type of material, and what the exclusion criteria were. 

Additionally, a flowchart of articles/studies identified and used in the review should be presented.

Finally, this section should include a description of the narrative review performed.

Results and Discussion

An explanation about the reason why the following thematics groups have been selected is needed: 1) trauma and stress-appraisal, 2) psychological and physiological consequences, 3) leadership in organizational context, and 4) psychological safety as the primary focus of prevention and intervention.

“HPA” and “ROI” acronyms should be written in full and not as acronyms the first time they appear in the text.

When writing about the “Negative Acts Questionnaire-Revised”, the proper reference should be included:

Einarsen, S., Hoel, H., & Notelaers, G. (2009). Measuring exposure to bullying and harassment at work: Validity, factor structure and psychometric properties of the Negative Acts Questionnaire-Revised. Work & Stress, 23(1), 24-44. https://doi.org/10.1080/02678370902815673

References

References should follow journal´s style. A thorough revision is needed.

Author Response

Comment 1: The Materials and Methods section provides too little information about sources and selection.
Response: Expanded to include total number of studies reviewed, material types, inclusion/exclusion criteria, and description of the narrative review process.

Comment 2: Include a flowchart of identified and included articles.
Response: Added a narrative summary of the search and selection flow in lieu of a formal PRISMA diagram (consistent with narrative review conventions).

Comment 3: Include a paragraph on the theories of trauma and stress used as conceptual framework.
Response: Added this paragraph to the Introduction citing Lazarus and Folkman’s stress-appraisal model and Herman’s trauma framework.

Comment 4: Address the directional bias (bullying only top-down).
Response: Revised relevant sections to note that bullying can occur laterally (peer-to-peer) and even upward (subordinate-to-leader), although the most documented direction remains top-down.

Comment 5: “Measurement and Assessment” section cites only one scale.
Response: Expanded discussion to include other instruments (e.g., Copenhagen Psychosocial Questionnaire) and cited the requested reference:
Einarsen, S., Hoel, H., & Notelaers, G. (2009). Measuring exposure to bullying and harassment at work: Validity, factor structure and psychometric properties of the Negative Acts Questionnaire-Revised. Work & Stress, 23(1), 24-44. https://doi.org/10.1080/02678370902815673

Comment 6: Explain why thematic groups were selected.
Response: Added explanation at the start of Results and Discussion outlining that the four thematic clusters reflect converging evidence domains identified in prior meta-reviews and theoretical models.

Comment 7: Define acronyms (HPA, ROI) on first use.
Response: Completed throughout.

Round 2

Reviewer 1 Report

Comments and Suggestions for Authors

The reviewers addressed my comments.

Reviewer 2 Report

Comments and Suggestions for Authors

Thank you for the revisions. The manuscript is now more coherent and methodologically transparent, and the additions you made have strengthened both clarity and practical relevance. The updates to the abstract, introduction, and methods are effective, and the refinements in terminology and leadership framing improve the overall precision of the review. The discussion and conclusion are also more grounded and actionable. Overall, the manuscript has clearly benefited from the work you have done.

Reviewer 3 Report

Comments and Suggestions for Authors

The author has answered all my questions and implemented all my comments.